

# Light alters the impacts of nitrogen and foliar pathogens on the performance of early successional tree seedlings

Alexander Brown[1,2] and Robert W. Heckman[1,3]

[1] Department of Biology, University of North Carolina at Chapel Hill, Chapel Hill, NC, United States of America
[2] Curriculum for the Environment and Ecology, University of North Carolina at Chapel Hill, Chapel Hill, NC, United States of America
[3] Department of Integrative Biology, University of Texas at Austin, Austin, TX, United States of America

## ABSTRACT

Light limitation is a major driver of succession and an important determinant of the performance of shade-intolerant tree seedlings. Shade intolerance may result from a resource allocation strategy characterized by rapid growth and high metabolic costs, which may make shade-intolerant species particularly sensitive to nutrient limitation and pathogen pressure. In this study, we evaluated the degree to which nitrogen availability and fungal pathogen pressure interact to influence plant performance across different light environments. To test this, we manipulated nitrogen availability (high, low) and access by foliar fungal pathogens (sprayed with fungicide, unsprayed) to seedlings of the shade-intolerant tree, *Liquidambar styraciflua*, growing at low and high light availability, from forest understory to adjacent old field. Foliar fungal damage varied with light and nitrogen availability; in low light, increasing nitrogen availability tripled foliar damage, suggesting that increased nutrient availability in low light makes plants more susceptible to disease. Despite higher foliar damage under low light, spraying fungicide to exclude pathogens promoted 14% greater plant height only under high light conditions. Thus, although nitrogen availability and pathogen pressure each influenced aspects of plant performance, these effects were context dependent and overwhelmed by light limitation. This suggests that failure of shade-intolerant species to invade closed-canopy forest can be explained by light limitation alone.

# INTRODUCTION

Shade-intolerant species are, by definition, unable to persist in low-light environments. This may occur because shade-intolerant species tend to allocate resources toward rapid growth and limited defense (*Walters & Reich, 1999*; *Myers & Kitajima, 2007*; *Valladares & Niinemets, 2008*). Because this shade-intolerant strategy prioritizes growth of new tissue over defense against consumers, the performance and survival of shade-intolerant plants may decline in environments where growth is slowed by nutrient or light limitation. Importantly, in these resource-limited environments, performance and survival can be

Corresponding author
Robert W. Heckman,
robert.heckman@utexas.edu

further reduced by high consumer pressure (*Coley, Bryant & Chapin, 1985*; *Fine, Mesones & Coley , 2004*; *Hahn & Maron, 2016*). In this way, consumer pressure may interact with light and nutrient availability to drive succession from high-light, early-successional fields to low-light, later-successional forests (*Augspurger, 1984b*; *Coley, Bryant & Chapin, 1985*; *Myers & Kitajima, 2007*; *Pasquini, Wright & Santiago, 2015*; *Griffin et al., 2016*).

The impacts of consumers on shade-intolerant species may increase as light and nutrients decline. This is because the fast-growing, poorly defended strategy is only advantageous when resources are ample enough to support high growth rates. Otherwise, these plants cannot maintain their high metabolic rates. By consuming plant tissue, pathogens and herbivores further reduce the ability of these plants to acquire the resources (e.g., light, nutrients, water) necessary for the growth and maintenance of their metabolically costly tissue, potentially driving precipitous declines in performance. Thus, in low-light conditions, shade-intolerant species may be especially susceptible to disease for several reasons. First, shade-intolerant species often exhibit low constitutive defenses (*Stamp, 2003*). Second, limitation by nutrients and light may prevent plants from constructing some defense compounds (*Hanssen et al., 2020*; *Huang, et al, 2020*). Third, light limitation may lead to the down-regulation of both the salicylic and jasmonic acid pathways (*De Wit et al., 2013*), which are key to responding to attack by pathogens and herbivores (*Thaler, Humphrey & Whiteman, 2012*; *De Wit et al., 2013*; *Ballaré & Pierik, 2017*). Moreover, nutrient limitation may be especially detrimental for shade-intolerant species (e.g., *Ward, 2020*), because it can drastically reduce their ability to build photosynthetic machinery and some defensive compounds (*Coley, Bryant & Chapin, 1985*; *Stamp, 2003*; *Hanssen et al., 2020*). Thus, susceptibility to pathogens among shade-intolerant species may be particularly high when light and nutrients are limiting (*Kitajima & Poorter, 2010*; *Griffin et al., 2016*; *Griffin et al., 2017*).

Regardless of resource availability, pathogens are critical drivers of dynamics in plant communities (*Mordecai, 2011*). Throughout forest and grassland systems, pathogens can limit seedling survival (*Hersh, Vilgalys & Clark, 2012*), ecosystem productivity (*Mitchell, 2003*; *Maron et al., 2011*), species' ranges (*Spear, Coley & Kursar, 2015*; *Bruns, Antonovics & Hood, 2019*), and can promote diversity (*Bever, Mangan & Alexander, 2015*; *LaManna et al., 2017*). Pathogens can alter community composition through negative density-dependent seedling mortality (*Comita et al., 2014*; *Bayandala, Masaka & Seiwa, 2017*; *Uricchio et al., 2019*; *Jia et al, 2020*), benefiting more resistant species (*Welsh, Cronin & Mitchell, 2016*; *Cappelli et al., 2020*), and those that have escaped their specialist herbivores and pathogens (*Heckman, Wright & Mitchell, 2016*; *Heckman et al., 2017*). Because pathogen impacts can also change with light and nutrient availability (e.g., *Dordas, 2009*; *Veresoglou et al., 2013*; *Heckman, Wright & Mitchell, 2016*; *Ballaré & Pierik, 2017*; *Liu et al., 2017*; *Agrawal, 2020*), shade-intolerant species may experience large differences in pathogen impacts across the range of habitats they occupy (*Augspurger, 1984a*). Thus, pathogens may reinforce shade-tolerance differences among species, promoting niche differentiation (*McCarthy-Neumann & Kobe, 2008*; *Krishnadas & Comita, 2018*).

In this study, we assessed the role of light and nitrogen supply in seedling susceptibility to pathogens and how  pathogen impacts on seedling performance are mediated by light and

nitrogen. We did this using an important pioneer species of old fields and early-successional forests, *Liquidambar styraciflua* (*Oosting, 1942*; *Wright & Fridley, 2010*; *Fridley & Wright, 2018*). In these early-successional environments, *L. styraciflua* can experience severe foliar disease (*McElrone et al., 2010*) and varied nutrient and light conditions. Thus, the interaction between pathogens and resource supply may be important for understanding successional dynamics. To date, this interaction has been addressed in only a few studies in natural systems (e.g., *Griffin et al., 2016*; *Griffin et al., 2017*; *Ward, 2020*). We predicted that:

1. Foliar fungal pathogen damage will be highest when light and nitrogen are both limiting.
2. Pathogen impacts on seedling height, a proxy for plant performance, will be highest when light and nitrogen are both limiting.

## METHODS

### Study system

*Liquidambar styraciflua* is a shade-intolerant deciduous tree that is common throughout the southeastern US. *L. styraciflua* is a key transitional species during succession—it competes well in early successional systems, but becomes less common as seedlings become increasingly shaded during succession (*Clark, LaDeau & Ibanez, 2004*; *Wright & Fridley, 2010*; *Hersh, Vilgalys & Clark, 2012*; *Addington et al., 2015*; *Brown et al., 2020*).

### Seedling propagation

We purchased *L. styraciflua* seeds from Sheffield's Seed Co. (Locke, NY). In the greenhouse at the University of North Carolina at Chapel Hill, we sowed seeds into flats. Ten days after germinating, each seedling was transplanted into a 2.84 L pot filled with 3:1 mix of potting mix (Fafard 3B; Sun Gro) and sterilized sand. To ensure that other soil nutrients would not limit seedling growth, the potting medium included 10 g P $m^{-2}$ as triple super phosphate, 10 g K $m^{-2}$ as potassium sulfate, and 100 g $m^{-2}$ micronutrients (Scotts Micromax, Marysville, OH), corresponding to 1 g triple super phosphate $plant^{-1}$, 0.45 g potash $plant^{-1}$, and 2 g micronutrients $plant^{-1}$ (*Borer et al., 2014*). On July 17, 2014, seedlings were moved to the field.

### Site description

We performed this experiment in an old field and adjacent forest in the Duke Forest Teaching and Research Laboratory, (Orange Co., NC). The old field has been maintained since 1996 through annual mowing. In the old field, *L. styraciflua* occurs as seedlings and small saplings. The adjacent forest is ∼40 years old and dominated by early successional trees such as *Pinus taeda*, *Liriodendron tulipifera* and *L. styraciflua*. Later successional species like *Acer rubrum* and *Quercus* spp. also occur throughout the forest.

We conducted this field experiment between July 17 and October 3, 2014 (11 weeks) using a split-plot design. At the whole plot level, we manipulated light availability; at the subplot level, we manipulated nitrogen availability and foliar fungal pressure. Each subplot was a single sweetgum seedling grown in its own pot; each whole plot was a cluster of four pots surrounded with a wire cage to exclude deer.
## Light availability

At the whole plot level, we assigned seedlings to levels of light availability using a replicated regression approach (*Cottingham, Lennon & Brown, 2005*). This entailed high replication (10×) at the two extremes—closed canopy (low light) and open field (high light)—and lower replication (4×) at three points along a transect from low to high light—at the forest edge, ~5 m from the forest edge, and ~10 m from the forest edge. Replicates were spaced ~5 m apart.

To assess light differences among treatments, we used Onset HOBO pendant light loggers (Onset Computer Corporation, Bourne, MA, USA). Loggers recorded light availability every 5 min for 10 days in early October. Because overstory trees had not yet begun to noticeably senesce and no disturbances (e.g., tree falls) had occurred, light availability in October should reflect relative light availability throughout the experiment.

## Nitrogen supply

Seedlings received five applications of aqueous ammonium nitrate ($NH_4^+NO_3^-$) over ten weeks (July 24–September 18). Seedlings under high nitrogen received 2 g N m$^{-2}$ application$^{-1}$ (in total, 10 g N m$^{-2}$ or 460 mg $NH_4^+NO_3^-$ plant$^{-1}$); seedlings under low nitrogen received 0.2 g N m$^{-2}$ application$^{-1}$ (in total, 1 g N m$^{-2}$ or 46 mg $NH_4^+NO_3^-$ plant$^{-1}$). In previous studies, the high nitrogen application rate increased soil nitrogen (*Stevens et al., 2015*) and alleviated nitrogen limitation at this site (*Fay et al., 2015*).

## Fungal pathogen pressure

Seedlings were either sprayed biweekly with a foliar fungicide or left unsprayed. This fungicide, Mancozeb (Dithane DF, Dow AgroSciences, Indianapolis, IN), was applied in late morning until it began to run off leaves. Mancozeb is a broad-spectrum non-systemic fungicide that has no known direct effects on photosynthesis, leaf longevity, shoot growth, or root growth (*Lorenz & Cothren, 1989*; *Kope & Trotter, 1998*; *Parker & Gilbert, 2007*), nor does it affect mycorrhizal fungi when applied at recommended rates (*Parker & Gilbert, 2007*). In a separate greenhouse study, fungicide reduced total biomass by ~10%, but this effect was only marginally significant ($P = 0.084$; Table S1A; Fig. S1A, Supplementary Methods 1).

In total, this experiment comprised 32 whole plots (10 high light, 10 low light, 4 at each of 3 points along a light transect; Fig. S3). Within each whole plot, there were 4 seedlings (2 nitrogen ×2 fungicide treatments), each growing in a separate pot, for a total of 128 seedlings.

## Measurements

In this study, we measured two responses—foliar damage and plant height—to determine whether light and nitrogen availability alter pathogen impacts on seedlings. Foliar pathogen damage was quantified visually by referring to digitized images of known damage severity (*James, 1971*; *Mitchell, Tilman & Groth, 2002*; *Mitchell et al., 2003*). We measured foliar damage as the percent of leaf area visibly damaged on October 3, 2014 (after 11 weeks in the field) on up to five leaves per plant, including the youngest and oldest leaves as well as three leaves evenly spaced in age. We measured damage as 0%, 0.1%, 0.5%, by

1% increments between 1 and 15% damage, then by 5% increments above this. Surveying leaves of different ages should best describe the mean level of damage across the entire plant, because damage typically increases with leaf age (*Hatcher, Ayres & Paul, 1995*; *Halliday, Umbanhowar & Mitchell, 2017*; *Heckman, Halliday & Mitchell, 2019*). Plant height was measured biweekly from July 25 until October 3, 2014 (6 observations over 73 days). Each time, we measured seedling height from the base through the end of the petiole of the highest leaf, which reflects the highest point at which seedlings can photosynthesize.

### Data analysis

We analyzed these data with linear mixed models in the nlme package (*Pinheiro et al, 2016*) in R version 3.5.3 (*R Core Team. 2019*). In all models, light was a categorical whole plot effect. Because both maximal and total daily light availability were similar for transect and high light seedlings ($P = 0.18$; Fig. S2), we combined these light treatments for all analyses. Thus, 88 seedlings from 22 whole plots were treated as high light and 40 seedlings from 10 whole plots were treated as low light. Nitrogen and spraying treatments were categorical subplot effects.

We quantified seedling height as the area under the curve of biweekly height measurements using the 'auc' function in the MESS package (*Ekstrøm, 2016*). To meet the normality assumption for linear models, foliar fungal damage was cubed-root transformed. This transformation best met the normality assumption because it is more strongly normalizing than the square root transformation, but less so than the log transformation. To reduce heteroscedasticity of height residuals, we used the varIdent function in 'lme' to allow variances to differ between light treatments (*Zuur et al , 2009*; *Pinheiro et al, 2016*). Post-hoc Tukey HSD tests were performed using the 'emmeans' function (*Lenth, 2018*).

## RESULTS

### Impacts of light and nitrogen on fungicide efficacy

Averaged across all light and nitrogen treatments, fungicide reduced fungal damage by 83% (Spraying, $P < 0.001$; Fig. 1A). Light and nitrogen availability jointly altered the effect of fungicide on visible foliar fungal damage (Light $\times$ Nitrogen $\times$ Fungicide; $P = 0.03$; Table S2): in high light, spraying did not alter fungal damage in either nitrogen level (Tukey HSD: High nitrogen, $P = 0.99$; Low nitrogen, $P = 0.83$); in low light, spraying reduced fungal damage to near zero at both nitrogen levels (Tukey HSD: High nitrogen, $P < 0.001$; Low nitrogen, $P = 0.001$).

### Impacts of light and nitrogen on foliar fungal damage

Among unsprayed seedlings, there were no significant main effects of nitrogen or light on foliar fungal damage (Nitrogen: $P = 0.5$; Light, $P = 0.11$; Table S3). Instead, light availability altered the effect of nitrogen on foliar fungal damage (Nitrogen $\times$ Light, $P = 0.01$; Table S3 ; Fig. 1B): in high light, fungal damage did not differ between nitrogen treatments (Tukey HSD: $P = 0.078$); in low light, fungal damage was over $3\times$ higher on high nitrogen than low nitrogen seedlings (Tukey HSD: $P = 0.039$). This is contrary to our prediction that damage would be highest under low light and low nitrogen.

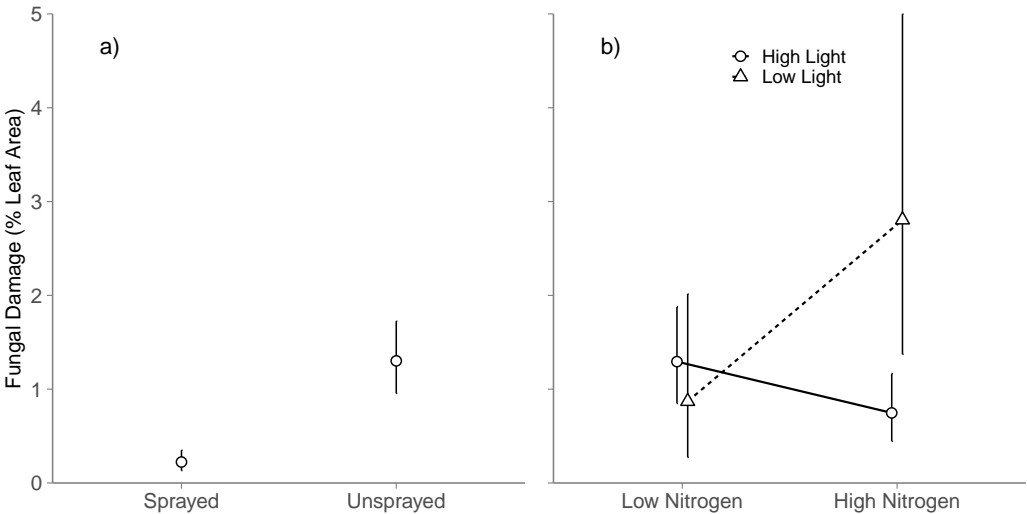

**Figure 1** **Effects of nitrogen and light availability on mean foliar fungal damage.** (A) Effects of fungicide spraying on mean foliar fungal damage ($N = 128$ seedlings), (B) effects of nitrogen and light availability on mean foliar fungal damage for seedlings not sprayed with fungicide ($N = 64$ seedlings: 20 in low light, 44 in high light); dashed lines represent plants growing in low light and solid lines represent plants growing in high light. Means and 95% confidence intervals were calculated using linear mixed models and back transformed from a cubed root transformation.

## Impacts of light, nitrogen, and fungicide on seedling height

Although light and nitrogen each had significant main effects on seeding height (Light, $P < 0.001$; Nitrogen, $P = 0.022$; Table S4), these two factors interacted to influence seedling height (Nitrogen × Light, $P = 0.004$; Table S4). In high light, seedlings grew 17% taller in the high nitrogen treatment than the low nitrogen treatment (Tukey HSD: $P < 0.001$). Height did not differ among nitrogen treatments in low light (Tukey HSD: $P = 0.62$).

Fungicide did not have a significant main effect on seedling height ($P = 0.14$). Instead, light availability altered the effect of fungicide on seedling height (Fungicide × Light, $P = 0.002$; Table S4 ; Fig. 2): in high light, sprayed seedlings grew 14% taller than unsprayed seedlings, but did not differ in low light (Tukey HSD: High light, $P = 0.002$; Low light, $P = 0.22$). Thus, although fungicide did not reduce visible damage, fungicide appears to have alleviated the negative impacts of fungal disease on seedling performance. Contrary to expectation, light and nitrogen did not interact with fungicide to influence seedling height (Nitrogen × Fungicide × Light: $P = 0.72$; Table S4).

## DISCUSSION

In this study, light, nitrogen, and pathogens additively and interactively influenced seedling performance. Contrary to our expectation, nitrogen and light availability interacted to alter foliar fungal damage: damage increased with increasing nitrogen availability, but only in low light. This suggests that increased nitrogen availability under low light makes plants more susceptible to enemies (*Dordas, 2009*; *Zhou et al, 2015*; *Ballaré & Pierik, 2017*). Although foliar damage was lower under high light, seedling height showed a different pattern:

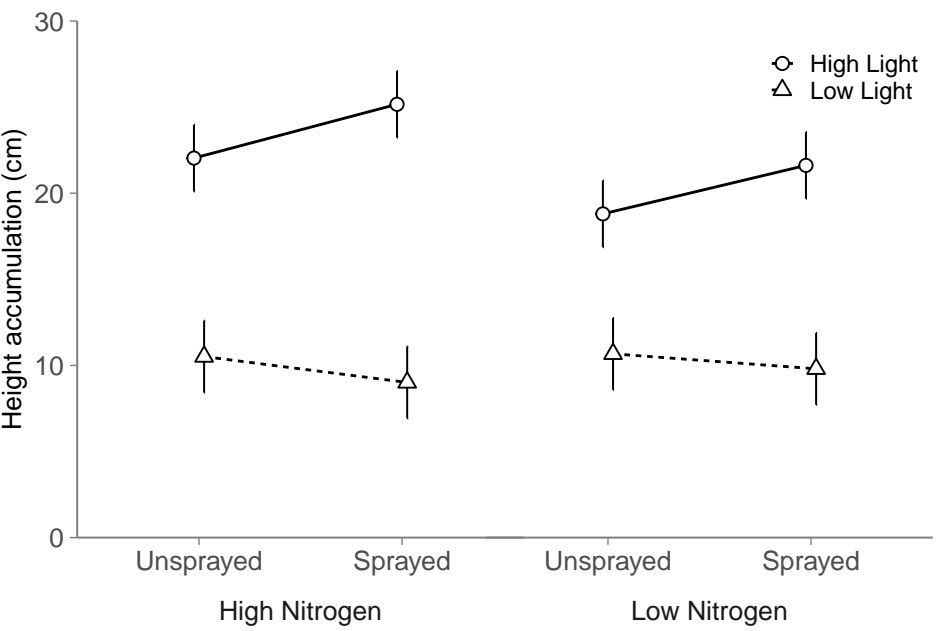

**Figure 2 Effects of nitrogen and light availability, and damage treatment on height accumulation in the field.** Effects of nitrogen and light availability, and fungicide spraying on seedling height ($N = 128$ seedlings: 40 in low light, 88 in high light). Dashed lines represent plants growing in low light and solid lines represent plants growing in high light. Means and 95% confidence intervals were calculated using linear mixed models.

pathogens impacted seedling height more strongly under high light than low light. This is contrary to other studies showing either greater impacts of pathogens and herbivores in low light or no difference in impacts between light environments (e.g., *Augspurger, 1983*; *Myers & Kitajima, 2007*; *Bayandala, Masaka & Seiwa, 2017*). Together, these results demonstrate the importance of light and nitrogen for modulating pathogen impacts on seedling performance.

Nitrogen and light availability interacted to influence pathogen damage in this study. Specifically, pathogen damage increased with nitrogen availability in the shade, but not in the sun. Because there was not a strong main effect of light, it is unlikely that differences in pathogen inoculum between high and low light environments were entirely responsible for this. Instead, low light may have reduced the ability of seedlings to synthesize defense compounds, either because they lacked the carbon to do so or because the jasmonic and salicylic acid pathways were down-regulated (*Stamp, 2003*; *De Wit et al., 2013*; *Zhou et al, 2015*; *Ballaré & Pierik, 2017*; *Huang, et al, 2020*). Either of these mechanisms would reduce the ability of seedlings to resist pathogens, leaving their nitrogen-rich leaves more susceptible to pathogen infection. Together, these possible explanations for the interactive effect of light and nitrogen on pathogen damage have important implications for succession (*Griffin et al., 2016*; *Griffin et al., 2017*): high nitrogen supply may reinforce the dominance of shade-intolerant species in early stages of succession (i.e., high-light environments) by contributing to growth without increasing damage, but may hinder performance of

shade-intolerant species in later successional stages by increasing damage without increasing growth (*Reinhart et al., 2010*).

Nitrogen and light availability also interacted to influence seedling height, a key proxy for seedling performance. Here, seedling height increased with nitrogen availability in high light, but not in low light. This indicates that light was more limiting than nitrogen to growth in the forest understory, but that nitrogen was more limiting than light in the old field. Despite this, nitrogen and light did not simultaneously interact with fungicide application to influence seedling height. Instead, spraying increased seedling height only in high light, which was contrary to our prediction that pathogen impacts on seedling performance would be larger under low light (*Augspurger, 1983*; *Stamp, 2003*; *Myers & Kitajima, 2007*). This indicates that fungal pathogens were negatively impacting seedling performance in high light, even without visible differences in fungal damage. Additionally, fungal pathogens may not have impacted seedling height in low light because seedling growth was severely light limited regardless of damage.

Although this study demonstrates important impacts of light, nutrients, and pathogens on *L. styraciflua* performance, there are several limitations. First, this study was short. While this short duration highlights a critical life stage—survival and growth of establishing tree seedlings (*De Steven, 1991*; *Fridley & Wright, 2018*)—we cannot account for differences in overwinter survival or impacts of light, nutrients, and pathogens beyond this window. Moreover, we cannot account for differences in shade tolerance across life stages (*Valladares & Niinemets, 2008*; *Falster, Duursma & FitzJohn, 2018*). Second, by growing seedlings alone in pots, we eliminated interspecific competition and may have reduced rain splash dispersal of pathogens. Reduced competition may have been more important in herbaceous-dominated old fields (*Flory & Clay, 2010*; *Fridley & Wright, 2012*) than in the more sparsely vegetated forest understory. Thus, growing seedlings in pots versus directly in the ground may have had a larger impact on plant growth in the herbaceous-dominated old field than in the forest understory. Third, we did not compare *L. styraciflua* to any other shade-intolerant or shade-tolerant species (e.g., *Chou, Hedin & Pacala, 2018*).

These results suggest that pathogens, nitrogen, and light can be important drivers of succession from old fields to forests (*Wright & Fridley, 2010*; *Fridley & Wright, 2012*; *Meiners et al., 2015*). Under high light, foliar disease reduced the performance of *L. styraciflua*. This high pathogen impact may ultimately slow the growth and spread of *L. styraciflua* in old fields, potentially slowing the conversion of herbaceous-dominated old fields to early-successional forests (*Gill & Marks, 1991*). In closed-canopy forests, pathogens may also be important determinants of species occurrence patterns (*LaManna et al., 2017*). For instance, seedlings of shade-intolerant species, like *L. styraciflua*, do not aggregate as often in forest understories as shade-tolerant species, indicating conspecific negative density dependence (*Clark, LaDeau & Ibanez, 2004*), which may result from foliar (*Hersh, Vilgalys & Clark, 2012*) or belowground disease. This may prevent shade-intolerant species from maintaining large enough seedling and sapling populations to exploit infrequent tree falls and could result in their exclusion from forest understories (*O'Hanlon-Manners & Kotanen, 2004*; *Wulantuya et al., 2020*). Moreover, pathogen impacts on shade-intolerant species can increase in shaded habitats, potentially increasing niche differentiation between

shade-tolerant and -intolerant species (*McCarthy-Neumann & Kobe, 2008*). As in other studies, pathogens in this study were important in low light, where foliar disease increased with increasing nitrogen availability. But increased disease had a limited impact on seedling performance. Instead, light limitation was too severe, even when pathogen pressure and nitrogen limitation were alleviated, to allow rapid seedling growth.

## CONCLUSIONS

In conclusion, we found that nitrogen and light interact to impact fungal damage, with the highest levels of damage at high nitrogen and low light. Despite this, spraying fungicide impacted seedling height more under high light than low light, indicating that pathogen pressure was higher when light was abundant and that fungal pathogens exerted negative impacts on seedling performance beyond what was visible on leaves. Overall, though, light limitation had the largest influence on seedling performance, overwhelming both nitrogen limitation and pathogen pressure (*Ward, 2020*). Given this, it appears that the shade-intolerant species *L. styraciflua* can be excluded from later-successional habitats solely by reducing light availability—high pathogen pressure and nitrogen limitation may further promote this, but are not required.

## ACKNOWLEDGEMENTS

We thank AJ Brown, F Halliday, C Mitchell, K O'Keeffe, A Simha, J Umbanhowar, and M Welsh for discussion on previous versions of this manuscript. Thanks to F Halliday and C Mitchell for suggestions on the design of the experiment, and J Umbanhowar for statistical advice. A Hurlbert and J Coyle lent us experimental supplies. J Garzoni and T Hodges provided help and expertise for the greenhouse experiment. P Wilfahrt helped with experiment maintenance.

### Funding

This study was funded by a National Science Foundation Doctoral Dissertation Improvement Grant to RWH (NSF-DEB-1311289). UNC's Summer Undergraduate Research Fellowship provided summer funding to Alexander Brown. The funders had no role in study design, data collection and analysis, decision to publish, or preparation of the manuscript.

### Grant Disclosures

The following grant information was disclosed by the authors:
National Science Foundation Doctoral Dissertation Improvement Grant: NSF-DEB-1311289.
UNC's Summer Undergraduate Research Fellowship.

### Competing Interests

The authors declare there are no competing interests.

## Author Contributions

- Alexander Brown and Robert W. Heckman conceived and designed the experiments, performed the experiments, analyzed the data, prepared figures and/or tables, authored or reviewed drafts of the paper, and approved the final draft.

## Data Availability

The raw data are available in the Supplemental Files.

## Supplemental Information

Supplemental information for this article can be found online at http://dx.doi.org/10.7717/peerj.11587#supplemental-information.

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
