# Peer review of "Light alters the impacts of nitrogen and foliar pathogens on the performance of early successional tree seedlings"

_PeerJ, doi:10.7717/peerj.11587_

## Round 0.1 · original submission · Major Revisions

This is a resubmission of a previously submitted manuscript which I handled. The previous reviewers were invited but were unable to re-review so I sought fresh reviewers.

The reviewers identified a variety of concerns that should be addressed. In addition, the reviewers have various suggestions for improving the manuscript.

·

Basic reporting

The work “Light alters the impacts of nitrogen and foliar pathogens on the performance of early successional tree seedlings” contains interesting information on the early stage of seedling development. The effects of light, nitrogen supply and treatments limiting the influence of fungal pathogens were investigated. The work is well written and easy to read (although as a non-native speaker I do not assess language correctness).

Experimental design

The aims of the work are properly constructed and the methods used allow to verify the hypotheses presented.

Validity of the findings

My main objection pertains the method of determining the degree of leaf damage and the statistical analysis of this feature. The authors wrote that they followed the method of W.C. James (1971) on the basis of which a visual assessment of the degree of damage was made. This method, although it expresses damage as a percentage, does not result in a measurement (continuous scale), only the result is assigned to a specific ordinal category, i.e. the data are expressed in an ordinal scale. This has the consequence that the data cannot be analyzed by ANOVA, even if it has previously been transformed. If the percentage categories had the same ranges, ANOVA could be used, but James did not propose one. Nowadays, when we have computers at our disposal, we perform this type of research with the use of image processing software. In this case, as the research has already been performed, another method of data processing should be used, appropriate for data expressed on an ordinal scale, e.g. a generalized linear model.

Additional comments

My other comments concern Supplementary materials.
Biomass results are reported, but it is not known whether they are in fresh or dry weight.
In the description of the S4 table, the term "day of clipping" appears, which is not mentioned in the current version of the work.

·

Basic reporting

The manuscript is dealing with a contemporary topic of ecological research. Study focusses on interactive impact of resource (light) and disturbance (fungal pathogen pressure) on growth performance of tree seedlings of Liquidambar styraciflua (a shade intolerant species). I think the study was conducted in a deciduous/ dry tropical environment. Measurement were taken continuously for the about 3 months. The data and results are substantial and would enrich the journal content after publication. However, the manuscript is not properly synthesized and the results are not properly discussed in light of literature available from the tree seedling ecology (See: Tripathi SN, Bhadouria R, Srivastava P, Singh R, Raghubanshi AS (2018) The effects of interacting gradient of irradiance and water on seedlings
of five tropical dry forest tree species, Tropical Ecology 59(3): 489-504) . However, even though this is written with good use of the English language, for which the author is to be commended. Thus, the manuscript needs a major revision before further proceedings. The comments are annotated in attached file.
• Add, practical implication of the study, in the last line of the abstract
• In introduction section, you may add some background from dry tropical/ deciduous forest to support your study.
• Citation of very old studies can be omitted by replacing new studies (delete at least 25% references)
• Discussion section can be reduced by 30% and should be based only on significant results of the study
• A separate conclusion section can be added
Thank you for giving me opportunity to review the manuscript.

Experimental design

The experimental layout should be added for easy understanding of the experimental design
Comments are annotated in the attached file

Validity of the findings

no comment

Additional comments

Comments are annotated in the attached file

Reviewer 3 ·

Basic reporting

Given the scarcity of information about the mechanisms by which pathogens may contribute to the maintenance of plant community diversity, the data presented in this manuscript have the potential to advance our understanding of plant-pathogen interactions in natural systems. That being said, there are substantial issues with the manuscript in its current state.

The article does not meet standards for clear and unambiguous, professional English. Throughout the manuscript, the following issues exist:
Some of the language is misleading.
For example, L10 states that L. styraciflua seedlings were grown “across a light gradient,” which is misleading because L176-179 state that light levels did not significantly differ between the “high light” and “transect”/intermediate light plots, so all of those data were considered high light in the analyses (i.e., light levels were either high or low in the analyses).
Related the legend for Fig. 1, reports the sample size as N = 32 whole plots and 64 subplots. This is misleading because, while this is not explicitly stated, it appears (based on L176-179) that 88 seedlings were treated as high-light seedlings in the analyses, while only 40 seedlings were treated as low-light seedlings.

Excessive and redundant text. Here are a few examples:
L3: “Shade intolerance may” would be more concise than “This shade tolerance habit may.”
L10: “for approximately three months” can be omitted.
L27: “also may depend critically on” can be shortened.
Paragraphs 1 and 2 of the Introduction are mostly redundant, and should be combined into a single impactful paragraph.
L43-45 could be better organized to eliminate unnecessary and redundant text. For example, “pathogens can limit seedling survival (CITATION), ecosystem productivity (CITATIONS) and species’ ranges (CITATIONS), and can promote diversity (CITATIONS).”
L97-98: Remove all three occurrences of “light” from the parenthetical. It is clear that you are referring to light.
L205: Rearrange the parenthetical so the Tukey HSD is only mentioned once.
L230: “there is a biologically…enemy damage” is unnecessary.
L281-283: Rearrange the sentence so that “field and greenhouse” is only used once.

Much of the text needs to be clearer and more explicit, and terminology needs to be consistent. Here are some examples:
L47: “survival” should be “mortality”
L61-62: “may lose more tissue to disease” than what? Shade-tolerant plant species or shade-intolerant plants in high light?
L62-64: Please clearly state why seedlings would experience more disease when growth is limited. Are you implying that the seedlings cannot escape disease fast enough (e.g., cannot lignify fast enough)?
L68-L69: “susceptibility of SEEDLINGS of an early…. impacted SEEDLING performance”
L77-78: Because survival is the most common metric for seedling performance, prediction 2 would be much clearer with text like this, “impacts on seedling height, a proxy for plant performance, will be.”
L91: Please state why shade-intolerant sweetgum saplings become more abundant near the forest’s edge. I presume this has to do with seed sources and dispersal limitation.
L122-123: Please use consistent and clear terminology for the various treatments used. Specifically related to fungicide application, rather than “enemy damage (high or low)” use “sprayed with fungicide and unsprayed.”
L134 and 136: It would be helpful if the number of seedlings in each treatment was clearly stated (e.g., N = 64).
L155: “impacts on SEEDLINGS”. Try to be more specific throughout the manuscript.
L158: It would be helpful if you stated how much time had elapsed since the seedlings were placed in their treatments (3 weeks). Did any foliar fungal damage occur in the greenhouse? If so, how was this accounted for in the analyses?
L158: Revise language for clarity. “UP TO five leaves per plant”
Details about the Tukey tests are not provided in the Data Analysis section.
L187-190 and the rest of the results section: change “spraying” to “fungicide”. Make sure that each section is interpretable on its own.
L220: Fix the sentence transition because the previous sentence is not fully “in conflict” with the cited studies.
L225: Change “Increasing” to “increased” since you only had 2 nitrogen treatments (high and low).
L287-290: This argument is really hard to follow, in part because of grammatical issues. I had to read it 3 times. Please revise it for clarity and consider splitting the ideas into more than 1 sentence.
L293: “immediate damage” is odd. Do you mean aboveground damage?
L303: Throughout the manuscript, consider using “former pasture” instead of “old field” for the high light treatment. Currently that term is confusing given you also make broader comparisons between the field and greenhouse (e.g., L282-283 and 288).
L322-323: change “plant: to “seedling”

Throughout the manuscript, please improve sentence structure and length, and other general grammatical issues. Here are some examples:
All compound adjectives require hyphens. For example, shade intolerant trees (e.g., L3,5,9) should be shade-intolerant trees (like in L51). Fix throughout the manuscript. Also fix: L38-39 “low nutrient” and L46 “negative density dependent”.
L85: To avoid informal writing, consider “less common as seedlings become increasingly shaded during succession.”
L216-217: This sentence seems to be missing punctuation and a word. “availability, BUT only”
L287: “had also had”

The Introduction, Discussion, and References do not provide sufficient background or context to demonstrate how the work fits into the broader field of knowledge. Here are some examples of relevant concepts and literature that are not, but should be, referenced in the Introduction and Discussion:
L21: Shade tolerance can vary across life stages for a given tree species.
The inhibition of defense compounds and pathogen resistance under low red:far‐red light ratio conditions should be addressed in the Introduction.
L22-23: The trade-off is not explicitly stated.
L52-53 should incorporate/reference Carol Augspurger’s research exploring pathogen impacts in the shaded understory vs. high light gaps.
L57: N limitation is not restricted to temperate forests. See LeBauer, D. S., & Treseder, K. K. (2008). Nitrogen limitation of net primary productivity in terrestrial ecosystems is globally distributed. Ecology, 89(2), 371-379.
L74-75: I am skeptical that this interaction has only been addressed in a few studies given the volume of agricultural literature focused on plant-pathogen interactions. Perhaps that statement is true for natural systems. Either way, the language needs to be modified.
While the manuscript is framed as an exploration of a potential mechanism contributing to succession, the Introduction and Discussion (e.g., in L244-253) should address the possibility that pathogens may reinforce shade tolerance differences among species, thereby enhancing niche differentiation. See McCarthy-Neumann, S., & Kobe, R. K. (2008). Tolerance of soil pathogens co‐varies with shade tolerance across species of tropical tree seedlings. Ecology, 89(7), 1883-1892.
The Discussion section needs to address an additional limitation of this study - that it only explores the effects of pathogens, nutrients, and light for a single shade-intolerant species. The study’s conclusions and central hypotheses about successional dynamics would be much more convincing with additional shade-intolerant species and shade-tolerant species. See Augspurger, C. K. (1984). Light requirements of neotropical tree seedlings: a comparative study of growth and survival. The Journal of Ecology, 777-795. She observed lower pathogen-caused mortality for seedlings of shade-tolerant species in the shade.
L218 needs additional text making a clearer link between the statement and the literature cited. L228 needs additional text making a clearer link between the statement and the literature cited.
L234: low R:FR compromises both salicylic acid‐ and jasmonic acid‐dependent pathogen defenses (de Wit, M. et al. 2013. The Plant Journal, 75: 90-103.)
L244-246: Different aggregation patterns could also be the result of dispersal. This should be addressed.
L244-246: What about the role of belowground pathogens in the observed patterns?
L277-281: The arguments made need additional clarification. “a larger impact on the results” needs to be clearer. On plant growth?
L312-313: Does Pinnus taeda experience relatively lower pathogen pressure? The argument made in L312-313 is incomplete.

While the structure of the article conforms to standard sections, the information contained within these sections, the subsections, and individual sentences needs to be organized in a more logical and readable manner.
The Introduction needs significant revision: improve the organization and clarity of ideas, eliminate redundant text, and improve the context for the study, which will require reviewing and incorporating additional pertinent literature.
The Methods section needs to be better organized. Here are some examples:
L125-127 is an example of a poorly organized, hard to follow sentence. Additionally, “three weeks before the end of the study” is out of place here.
Seedling propagation (L144-153) is near the end of this section, yet it occurred first in the study.
L167-171 Seems more appropriate for the Data Analysis subsection.
The Results and Discussion section would be much easier to follow if they clearly presented each main effect and then their interactions.
L225-230: Improve the readability of the paragraph by eliminating redundant text and rearranging the sentences.
L244-253 Seems better suited for somewhere near the end of the discussion.
L521: These details, “from 8 days after… (73 days),” in Figure 2 are absent from, and should be moved to, the Methods section.

The main text figures are relevant to the content of the article. However:
Fig. 1 should be a two-panel figure depicting fungal damage for seedlings subjected to BOTH fungicide treatments (sprayed and unsprayed), not just unsprayed. The results for the seedlings treated with fungicide should not be buried in the supplement.
The figure legends require revision - eliminate unnecessary details (see L521) and include meaningful sample sizes. Related, “seedlings” would be clearer than “subplots” (L518 and 524).

Experimental design

Yes, this original, primary research falls within Aims and Scope of PeerJ; however, some issues need to be addressed before I would consider the article to be scientifically and methodologically sound (described below).

I have concerns about maintaining seedlings in pots (L100). These seedlings would have been unnaturally suspended above the soil and similar-sized vegetation, which has implications for their exposure to pathogens, particularly splash-dispersed pathogens. At the very least, this needs to be addressed within the Discussion section.

The research questions are reasonably well-defined, relevant & meaningful. It is less clear how this research fills the identified knowledge gap given that it only explores the effects of pathogens, nutrients, and light for a single shade-intolerant species, and to understand the mechanisms shaping succession and heightening niche differences among plant species it is necessary to evaluate additional shade-intolerant species and shade-tolerant species.

The Methods section requires revision. It is extremely hard to follow and there is missing information. These issues make it challenging to assess the scientific merit of the study and the validity of the findings.

Validity of the findings

I questioned the validity of the findings for a few reasons (described below), but I believe that the authors will be able to address my concerns in revision.
The seedlings were in pots (L100), as opposed to being planted in the ground. My concerns about this are described in the “Experimental design” section.
The authors intended to grow L. styraciflua seedlings across a light gradient; however, L176-179 state that light levels did not significantly differ between the “high light” and “transect”/intermediate light plots, so all of those data were considered high light in the analyses (i.e., light levels were either high or low in the analysis). This means 88 seedlings received the high light treatment, while only 40 seedlings received the low light treatment. How did this impact the results? This is not necessarily a disqualifier, but more transparency is needed within the manuscript.
It is unclear whether the statistics used are appropriate. For example, the authors cube root transformed their fungal damage data (L180). Because of the pitfalls associated with data transformation, there has been a shift from data transformation towards the use of non‐normal error structures and generalized linear models. Were generalized linear mixed-effects models considered/used? Were the results similar?
L142: There are separate root and shoot measurements in peerj-49499-Brown_and_Heckman_greenhouse_fungicide_data.csv. If the fungicide affects the root:shoot ratio (which it appears to do based on the root masses), there may be important implications for the plant performance (seedling height) metric.

---

## Round 0.2 · Minor Revisions

I appreciate your attention to revisions suggested by the three reviewers. I agree that the manuscript has been improved and I don't think this manuscript needs to be returned to the reviewers.

I have a number of minor comments and suggestions that mainly focus on increasing clarity of presentation or wording:

Specific commons
L 17-19. Consider condensing, perhaps simply: “This suggests that failure of shade-intolerant species to invade closed-canopy forest can be explained by light limitation alone.”
L 35 Check wording. Would this be clearer: “Otherwise, these plants cannot maintain their high metabolic rates”?
L 62 It is not clear how a shade-intolerant species could occupy a closed canopy forest. If it could, wouldn’t it be incorrect to call it a shade-intolerant species? Perhaps this could be reworded to avoid this apparent contradiction?
L 63 Do you mean light tolerance range?
L 75 It is correct to say that a “disease is highest…”? Perhaps “foliar pathogen damage is highest…” or “foliar impact of disease is highest…”
L 150 Can you briefly comment on how / whether non-pathogen foliar damage was considered in the leaf scoring process?
L 195-195 I didn’t follow this sentence. What does “This” refer to?
L 215 for clarity, I suggest “pathogen inoculum”
L221-222 Considering that leaves never seemed to have > 5% damage in any light treatment, it this a reasonable explanation? Was there any evidence of abscission?
L 240-243 I didn’t follow this reasoning. If both fungal and immune responses are visible, then why would plants with little or no visible damage (high light) respond to fungal treatment? Perhaps unsprayed high light plants are allocating resources to defenses? Spraying then allows these resources to be allocated towards growth? Aldea et al. 2006 also discuss the idea that plants may allocate resources to defense, which may reduce allocation to photosynthesis. Defense responses are not necessarily visible. Another example is closing of stomata, which would likely reduce growth rates.
L 255 seedling should say seedlings
L 256 plants growing in pots were also planted directly in soil (within the pots). It would be clearer to say something like “versus planting them directly in the ground…”
L 258-263 Since the greenhouse treatment, which was only briefly mentioned in Methods, did not significantly reduce plant growth, there would seem to be no point in comparing field and greenhouse effects of the fungicide here. These sentences seem like unnecessary speculation.
L267 This sentence does not seem to be accurate. Your study did not show the importance of any factors in succession. Doing that would require a longer term study observing succession. Instead, you could say your study results suggested the potential importance of x,y,z in succession, as supported by studies a,b,c.
L 270 “succession toward woody dominance slows…” – It’s not clear how your results support this statement. You did not study woody dominance during succession and furthermore, some woody species are shade-tolerant.
L 273-275 I think Brown et al explained the more clumped distribution of saplings in shade tolerant tree versus shade-intolerant trees as a function of their differing plant functional strategies, not in terms of different susceptibilities to pathogens, as implied in this sentence.
L 279 This could use brief explanation or rewording to make it understandable and relevant. I think a key finding of McCarthy-Neumann and Kobe that is most relevant to your study is that less shade-tolerant species tended to be more strongly affected by pathogens when growing in a shaded environment.
L 281 Why is there a colon connecting the two components of this sentence?
L 294 In place of exacerbate, I suggest “further promote”
Figure 2 – on my computer the Y-axis label says “cm 73” – what does this mean? I suppose it might be cm per 73 days which is a strange unit, but in that case the values shown are around 1000 cm, which amounts to growth of 10 meters in height, which does not seem feasible for your short study.
Figure 2 – why do all of the 95% confidence intervals on this graph appear to be exactly the same? Please check the values.

---

## Round 0.3 · accepted · Accept

The study presents the results of a straightforward field experiment. Limitations are appropriately acknowledged in the Discussion. The requested revisions have been made. I have one minor comment regarding height versus height accumulation (figure 2): Can you clarify in the caption of Figure 2 whether the y-axis “accumulation” refers to the change in height during the experiment or the total ending height? Based on the current wording of line 191, it seems that your comparison of % difference (14%) is based on the total height, referring to Fig 2, so I would expect the y-axis of Fig 2 to be “final height”. In my mind, it could be more insightful to report the magnitude of effect (% increase in growth) as a percentage of the change in height during the experiment rather than as a percentage of the final height, since the latter % is influenced by the starting height, which the reader cannot determine.